# Resistance and Susceptibility Immune Factors at Play during *Mycobacterium tuberculosis* Infection of Macrophages

**DOI:** 10.3390/pathogens11101153

**Published:** 2022-10-06

**Authors:** Jan D. Simper, Esteban Perez, Larry S. Schlesinger, Abul K. Azad

**Affiliations:** 1Host-Pathogen Interaction Program, Texas Biomedical Research Institute, 8715 W. Military Drive, San Antonio, TX 78227, USA; 2Department of Microbiology, Immunology and Molecular Genetics, UT Health Science Center San Antonio, San Antonio, TX 78229, USA; 3Translational Sciences Program, UT Health San Antonio Graduate School of Biomedical Sciences, San Antonio, TX 78229, USA

**Keywords:** tuberculosis, macrophages, mouse model, innate immunity, host defense, host genetic variation

## Abstract

Tuberculosis (TB), caused by infection with *Mycobacterium tuberculosis* (*M.tb*), is responsible for >1.5 million deaths worldwide annually. Innate immune cells, especially macrophages, are the first to encounter *M.tb*, and their response dictates the course of infection. During infection, macrophages exert a variety of immune factors involved in either controlling or promoting the growth of *M.tb*. Research on this topic has been performed in both in vitro and in vivo animal models with discrepant results in some cases based on the model of study. Herein, we review macrophage resistance and susceptibility immune factors, focusing primarily on recent advances in the field. We include macrophage cellular pathways, bioeffector proteins and molecules, cytokines and chemokines, associated microbiological factors and bacterial strains, and host genetic factors in innate immune genes. Recent advances in mechanisms underlying macrophage resistance and susceptibility factors will aid in the successful development of host-directed therapeutics, a topic emphasized throughout this review.

## 1. Introduction

*Mycobacterium tuberculosis* (*M.tb*) is an intracellular pathogen that has been causing disease in humans for thousands of years and continues to be among the most lethal infectious diseases today. Tuberculosis (TB) infection is caused by the inhalation of airborne droplets from individuals with active disease, although 90–95% of primary infections lead to asymptomatic control rather than progression to active disease [1]. This suggests the presence of innate immune factors that both respond to initial TB infection and reduce the risk of latent TB reactivation. Central to regulating these immune factors and response to *M.tb* are macrophages. Among the challenges that come with identifying such factors in macrophages include differences between animal models and human data. For example, the immune factor nitric oxide (NO) is critical for *M.tb* control in mice [2], but the early control of *M.tb* growth in human macrophages was shown to be NO-independent [3]. In this review, we will discuss similarities and differences between study models in each section as appropriate. We will focus this review on recent advances in our understanding of the innate immune factors that help control or promote *M.tb* infection by macrophages (summarized in Table 1) and how these may help define new host-directed therapeutic approaches. It is noteworthy that most of the work in this area is in in vitro tissue culture macrophage models and cell lines with limited attention to alveolar macrophages (AMs) which are the unique, first macrophage type that is infected by *M.tb* and initiates pathogenesis [4]. Aside from AMs, human blood monocyte-derived macrophages (MDMs) are the current best alternative primary human macrophage type to study, with the benefit of increased translation to human disease but with some differences in the expression of and response to immunological factors [5]. Cell lines commonly used in TB research are THP-1 and U937 cells, which are human monocytic cell lines isolated from leukemia patients and which can be differentiated into macrophage-like cells [6]. These are more distant models from human AMs but offer technical ease of use. Mouse cell lines include RAW 264.7 macrophages as perhaps the most commonly used model, although bone marrow-derived macrophages (BMDMs) and macrophages isolated from lung homogenates are also used. Finally, animal models include mice, rabbits, guinea pigs, zebrafish, and non-human primates, each with varying susceptibility to *M.tb* infection and translatability to human disease [7].

## 2. Brief Overview of *M.tb* Infection Pathogenesis

*M.tb* is an airborne infection transmitted by the inhalation of droplets from active TB patients. Bacteria in the droplets are deposited in the alveoli and encounter AMs, which phagocytose the bacteria via different cell surface receptors [4,70,71]. Receptor-mediated signaling and trafficking are critical in initiating the immune response, which either controls or promotes TB infection and subsequently shapes the development of the adaptive immune response [71,72]. Work continues to be performed on elucidating the intracellular mechanisms underlying the differences in *M.tb* control by macrophages.

## 3. Cellular Pathways

Apoptosis, or programmed cell death, is an important host defense mechanism against infection as it can promote pathogen clearance and resolution of inflammation [73]. *M.tb* has evolved various strategies for inhibition of apoptosis, ranging from the inhibition of calcium influx, suppressing cell death pathways, or promoting the expression of anti-apoptotic factors [74,75]. Our lab recently identified one such mechanism, in which *M.tb* induces the expression of the peroxisome proliferator-activated receptor (PPAR)γ in human MDMs, which in turn drives the expression of Mcl-1, a pro-survival member of the Bcl-2 family. Treatment with Mcl-1 antagonists resulted in significantly decreased *M.tb* growth in MDMs [8]. Another study described the critical role of caspase-8 in driving the cell death of *M.tb*-infected mouse macrophages and the control of infection in mice by enhancing apoptosis through the use of specific treatments [9]. Sirtuin 7 was another factor described to help control *M.tb* growth through NO-induced apoptosis in RAW 264.7 cells [10]. There has also been an increasing interest in studying the function of different microRNAs in the context of programmed cell death [76,77,78,79,80]. Other microbiological and host factors have also been described to attenuate the inflammatory response by inhibiting apoptosis [81,82]. Overall, there is strong evidence that apoptosis is an important process for *M.tb* control, and promoting the activity of this pathway as a viable host-directed therapeutic approach is worth further investigation.

Autophagy is an evolutionarily conserved process in eukaryotic cells in which intracellular components are transported to lysosomes for degradation and recycling. It is a core process for the maintenance of cellular and organismal homeostasis [83]. There is evidence that autophagy is an important process to control TB infection, and host-directed therapies to promote autophagy have become more popular [84]. Recent advancements include targeting the mammalian target of rapamycin (mTOR) signaling in mice [11], hydrogen sulfide in mouse RAW 264.7 macrophages [12], or hypoxia-inducible factor 1 (HIF-1) in human U937 monocytes [13] to increase autophagy and the control of *M.tb* growth in infected cells. It was also shown that the upregulation of DNA damage-regulated autophagy modulator 2 (DRAM2), which is thought to play a role in the initiation of autophagy, leads to decreased *M.tb* growth in human MDMs [14]. The upregulation of the microRNA miR-18a decreased LC3-II expression, a marker of autophagosome formation, in RAW 264.7 cells, and promoted *M.tb* survival [15]. Our collaborative studies have also shown that agonists of the C-type lectin receptor CLEC4E and Toll-like receptor TLR4 enhance autophagy and decrease *M.tb* growth in mouse bone marrow-derived macrophages (BMDMs) and lungs [16]. Finally, sirtuin 3 (SIRT3), an NAD+ dependent deacetylase, was shown to be important for the expression of PPARα, an activator of autophagy in mouse BMDMs, and *M.tb* grew significantly more in SIRT3^−/−^ BMDMs [17]. The induction of autophagy continues to show promise as an avenue for host-directed therapies against TB infection.

An inflammasome is a group of intracellular protein complexes that are critical in the innate immune system to respond to infection by activating caspase-1 to cleave pro-IL-1β/IL-18 which then go on to trigger downstream inflammatory pathways [85]. Multiple proteins can induce the formation of an inflammasome, such as NLRP3, AIM2, NLRC4, or NLRP1, although NLRP3 is perhaps the most characterized. Triggering the inflammasome by *M.tb* requires the ESX-1 type VII secretion system [86,87,88,89], and it has been suggested that this is due to plasma membrane damage mediated by ESX-1 [90]. There are conflicting reports of whether inflammasome induction is beneficial for the control of *M.tb* or whether too much of it can lead to worsened outcomes and an overly inflammatory response. A recent study showed a decreased rate of the early growth of *M.tb* in infected BMDMs from NLRP3^−/−^ mice, as well as from WT mice administered with an NLRP3 inhibitor; however, the ability of different *M.tb* strains to induce inflammasome formation was variable [91]. The inhibition of P2X7, a receptor that detects ATP released during cellular stress or death pathways and activates the inflammasome, was associated with decreased disease severity in mice and CFUs per lung [18]. However, another study found that P2RX7 potentiation through the drug clemastine improved *M. marinum* control in zebrafish [19]. This highlights the importance of differences in disease models during the study of TB, and more work is needed to elucidate the precise role of inflammasome signaling during *M.tb* infection, especially in human macrophages.

*M.tb* infection of human AMs and MDMs has been described to induce a shift from oxidative phosphorylation to aerobic glycolysis [92], and this shift has been found to be required for the effective control of bacterial growth [93]. Aerobic glycolysis promotes the activity of the transcription factor hypoxia-inducible factor 1α (HIF-1α), especially in IFN-γ-activated macrophages, and macrophages lacking HIF-1α are defective in the IFN-γ-mediated control of infection [93]. The treatment of human MDMs with a HIF-1α stabilizer at normoxic conditions decreased intracellular *M.tb* growth, although it also decreased the release of TNFα and IL-10 [20]. Mice with HIF-1α-deficient myeloid cells displayed increased lung bacterial burden compared to WT mice 120 days post-infection and when infected with hypervirulent *M.tb*, although this was not seen at lower dose infections [21]. A separate study found that BMDMs from mice lacking HIF-1α in their myeloid cell lineage had increased *M.tb* growth, and the expression of glycolysis-related genes was impaired by HIF-1α KO [22]. However, the chronic expression of HIF-1α can also lead to unwanted pathology, and a mechanism was described by which IL-17 negatively regulates HIF-1α to reduce lung bacterial burden and hypoxic granuloma formation in mice [94]. As with many innate immune factors in TB, a balance must be maintained between the immediate beneficial effects and harmful chronic complications.

## 4. Bioeffector Molecules

The first step in initiating an immune response to *M.tb* infection is the recognition of pathogen-associated molecular patterns (PAMPs) by pattern recognition receptors (PRRs). The PRRs, known as Toll-Like Receptors (TLRs), play an important role in recognizing *M.tb* ligands and polymorphisms in TLRs have been found to be associated with increased TB susceptibility [95]. Various studies have identified Toll-like receptor 2 (TLR2), which recognizes lipomannan and lipoprotein from *M.tb* [96,97], and TLR9, which recognizes unmethylated CpG motifs in bacterial DNA [25], as key TLRs for the recognition and control of *M.tb* infection in mouse BMDMs. TLR4 and TLR8 have also been studied in the context of TB infection [95]. It was recently discovered that the *M.tb* virulence factors phthiocerol dimycocerosates (PDIM) and ESAT-6 secretion system 1 (ESX-1) inhibit a late endosome-specific component of the TLR2 response to improve *M.tb* growth in mouse BMDMs [23]. TLR2 was also shown to be critical for the activation of SIRT3 and protection against *M.tb* in mouse lungs and spleen [24]. 

Indoleamine 2,3-dioxygenase-1 (IDO-1) is the rate-limiting enzyme involved in tryptophan metabolism to downstream metabolites such as kynurenine in host cells. It was found that *M.tb* infection induced the upregulation of IDO-1 expression in both human and murine macrophages, and infected macrophages produced an immunosuppressive metabolite (kynurenine) that did not reduce *M.tb* growth in vitro [26]. IDO-1 expression correlated with an increase in mouse lung CFUs and the treatment of rhesus macaques with an IDO-1 inhibitor decreased lung bacterial burden [27]. IDO-1 likely represents one of several biochemical pathways in macrophages that prevent the efficient killing of *M.tb* in TB granulomas [28]. IDO-1, and by extension, the kynurenine/tryptophan ratio, has therefore garnered interest as a possible biomarker for TB infection.

Reactive oxygen species (ROS) are highly reactive oxygen-containing molecules that can destroy both pathogenic bacteria as well as host cell machinery. As an example, NADPH oxidase catalyzes the transfer of electrons from NADPH to molecular oxygen to form superoxide [98]. Multiple other ROSs, such as hydrogen peroxide, are then produced and can react quickly with other molecules in their immediate cellular location. The overexpression of focal adhesion kinase (FAK) in THP-1 macrophages led to decreased *M.tb* survival, and this was due to increased ROS production [29]. The microRNA miR-495 was shown to decrease the survival of H_37_R_v_ in THP-1 cells through the increased production of ROS and the inhibition of SOD2 [30]. The knockdown of the receptor TARM-1 in RAW 264.7 cells decreased the production of ROS and increased the growth of *M.tb* H_37_R_v_ [31]. Additionally, the inhibition of fatty acid oxidase in mouse BMDMs led to NADPH oxidase recruitment and decreased *M.tb* growth [32]. Interestingly, the depletion of platelets in mice led to decreased lung CFUs by the increased production of ROSs [33]. The same study found that the treatment of human MDMs with soluble CD157 led to the decreased CFUs of *M.tb* and described a pathway by which CD157 participates in TLR2-dependent ROS production [33]. However, although ROSs may be beneficial if they target pathogens, they can also harm host cells and cause more inflammation. Protecting the host, therefore, requires a potential therapy, and one study found that the supplementation of liposomal glutathione, an antioxidant that prevents damage to immune cells by ROS, decreased lung CFUs in mice [34]. Overall, there is support for the importance of ROS generation in combatting TB infection.

Cohort studies have shown that a deficiency in vitamin D is associated with an increased risk of TB [99,100]. Activated vitamin D_3_ induces the synthesis of the cathelicidin antimicrobial peptide LL-37, which enhances xenophagy [35], and the administration of LL-37 in *M.tb*-infected mice led to a reduction in lung bacterial burden [36]. Different strains of *M.tb* also required different concentrations of LL-37 to inhibit their growth [101]. However, randomized controlled clinical trials have not shown a benefit of vitamin D supplementation in the prevention of disease [102,103], or that supplementation provided additional benefit to patients already receiving antibiotics [104,105]. A recent study found that vitamin D in conjunction with phenylbutyrate inhibited the growth of multi-drug resistant tuberculosis (MDR-TB) strains in human macrophages [37]. More work is needed to understand the precise role of vitamin D (potentially in combination with other micronutrients) in controlling *M.tb* growth in macrophages.

Peroxisome proliferator-activated receptors (PPARs), of which there are three in humans (PPARα, PPARβ/δ, and PPARγ), are members of a ligand-binding nuclear receptor family and regulate metabolic, differentiation, proliferation, and inflammatory pathways in their roles as transcription factors [106]. PPARα was demonstrated to be protective against infection since BMDMs from PPARα^−/−^ mice had increased *M.tb* growth compared to WT [38]. SIRT3 was also described to have anti-mycobacterial effects through PPARα [17]. Work from our lab has shown that PPARγ knockdown in human MDMs significantly decreases *M.tb* growth concomitant with an increase in TNF [39], suggesting differential roles within the PPAR family members in *M.tb* control. As mentioned earlier, we also showed that the inhibition of the PPARγ effector and anti-apoptotic protein Mcl-1 resulted in greatly decreased *M.tb* growth in MDMs [8]. It was also found that the bacteriostatic effect of vitamin B1 on *M.tb* growth in mouse BMDMs was PPARγ-dependent; treatment with a PPARγ agonist alone increased *M.tb* growth [107]. Additionally, a model has been suggested by which targeting PPARy leads to increased IL-1β, IL-12, and iNOS production and decreased IL-10 production to protect against *M.tb* infection in a murine model [108]. Overall, the evidence so far points to PPARα’s protective role and PPARγ’s permissive role in the growth of *M.tb* in macrophages.

## 5. Other Cellular Factors

The alveolar lumen is composed of airway epithelial cells (AECs), which are subdivided into alveolar type I cells and type II cells [109]. Type I cells are known for their involvement in gas exchange [109] while type II cells are involved in the production and recycling of lung surfactant [110]. AECs play an integral role in maintaining airway homeostasis and have the capacity to respond to changes in the external and internal environment via immunomodulatory secreted molecules [111]. Furthermore, these cells are known to express PRRs and surfactant which are important in the recognition of *M.tb* by lung cells [40]. An in vitro study showed that after exposure to *M.tb* or *M.tb*-infected macrophages, AECs promote the recruitment of neutrophils, offering a strong indication that they contribute to the protective host response against *M.tb* infection in the airway environment [41]. Additionally, a study using a lung-on-a-chip model with mouse cells showed that both alveolar epithelial cells and macrophages lacking surfactant allowed for the rapid growth of *M.tb*, further highlighting the importance of surfactant in bacterial control in an alveolar environment [42]. Finally, alveolar lining fluid, which bathes AMs, contains enzymes called hydrolases that release *M.tb* cell envelope fragments extracellularly which in turn improve the ability of macrophages to control *M.tb* growth by initiating a robust innate immune response [64].

## 6. Microbiological Factors

**ManLAM:***M.tb* has a complex cell envelope containing predominantly unique glycolipids and lipoglycans that confer a survival advantage for bacteria and that vary in strains of different lineages, potentially explaining differences in their virulence [112,113,114]. Among the most studied is mannose-capped lipoarabinomannan (ManLAM), a heterogeneous, amphipathic lipoglycan that serves as both an immunogen and virulence factor for *M.tb* [115,116]. It is composed of a phosphatidyl-myo-inositol (PI) anchor, a carbohydrate core, and various mannose-capping motifs [117]. The recognition of ManLAM in the lipid-rich cell wall of *M.tb* by the macrophage mannose receptor (MR, CD206) leads to a pathway of the intracellular survival of bacteria within the host by blocking phagosome fusion with lysosomes [118,119]. Binding to MR also produces an anti-inflammatory response as measured by the production of IL-10 and TGF-β while limiting the production of pro-inflammatory cytokines such as TNF, IL-6, IL-1β, and IL-12 [120,121]. Additionally, ManLAM binding to MR enhances the expression of the nuclear receptor PPARγ and the signal transducer and activator of transcription (STAT)-5α [39]. An aptamer that inhibited *M.tb* entry into murine macrophages by targeting ManLAM led to increased IL-1β and IL-12 production but decreased IL-10 production [108]. More work needs to be performed for elucidating the intracellular signaling pathways triggered by ManLAM and their potential as therapeutic targets for TB infection.

**ESAT-6/ESX-1:** ESAT-6 secretion system 1 (ESX-1) is a type VII secretion system and a major *M.tb* virulence factor, with host effects ranging from the induction of necrosis, autophagy, NOD2 signaling, and type I interferon production [122]. Additionally, the perforation of phagosomes in macrophages is mediated by *M.tb* ESX-1 through ESAT-6 [123], thereby enhancing *M.tb* virulence via the release of *M.tb* molecules into the cytosol [124] or by the translocation of *M.tb* into the cytosol [123]. Cytosolic translocation was found to be a feature of only virulent mycobacterial strains [123]. To further enhance virulence, *M.tb* strain H_37_R_v_ suppresses the apoptosis of macrophages by acting on members of the anti-apoptotic Bcl-2 family, especially Mcl-1 [8,125]. Additionally, this strain funnels cell death down the pathway of the necrosis of infected neutrophils in an ESAT-6-dependent manner, a process that promotes bacterial growth following uptake by macrophages [126]. Because of its strong virulence and potent antigenic properties, ESAT-6 is being used for the development of a new preventive and therapeutic vaccine for TB [127]. 

## 7. Mycobacterial Strains

There are many adaptations that *M.tb* has evolved over the course of its history, and the strains can differ from one another in their beneficial mutations. The *M.tb* Rv1096 strain was recently found to accelerate its growth in RAW 264.7 cells and dampen pro-inflammatory cytokine production as well as NF-κB and MAPK signaling [128]. The *M.tb* protein Rv0455c was shown to be a virulence factor as a Δrv0455c mutant had impaired growth in lung and spleen homogenates due to decreased iron uptake [129]. Different strains may induce different immune cell and cytokine responses, such as IL-22 being important for protective immunity against the hypervirulent HN878 strain but not against the lesser virulent H_37_R_v_ strain [130]. In a study on the interplay between different *M.tb* strains and races/ethnicities, strain CDC1551 elicited lower levels of IL-1, IL-6, IL-10, TNF-α, and GM-CSF production from human MDMs and higher levels of IL-8, compared to H_37_R_v_ and HN878 [131]. Factors that are not important for the control of one *M.tb* strain may have a more important role for a different strain, such as having been recently shown with CCR2^−/−^ mice being more susceptible to HN878 compared to H_37_R_v_ [132]. Infection with the *M.tb* Erdman strain led to a higher lung bacterial burden and a greater systemic inflammation due to increased response to hypoxia compared to CDC1551 in a non-human primate model [133]. It is evident that different strains induce differential host responses mediated in large part by macrophages; therefore, the development of future host-directed therapies will need to take these differences into account to achieve the most success.

## 8. Other Microbiological Factors

Much work is being performed to identify important genes and factors for *M.tb* virulence and survival. The knockout of the Rv2617c gene in the CDC1551 strain reduced its growth in mice and decreased oxidative stress resistance [134]. The *M.tb* PE_PGRS20 and PE_PGRS47 proteins interact with the Ras-related protein Rab1A to prevent autophagy in infected host cells and infection of THP-1 macrophages with deletion mutants showed decreased growth [135]. The balance of micronutrients is important for *M.tb* survival. The bacterial ATPase CtpB was found to be important for the regulation of copper levels and the optimal growth of *M.tb* both in vitro and in the mouse model [136]. The ability of *M.tb* to mutate and overcome the loss of the ESX-3 type VII secretion system, involved in iron acquisition, and thus restore virulence, has also been described in the mouse model [137]. Interestingly, the aggregation of *M.tb* bacilli was described to cause earlier pro-inflammatory gene activation and cell death compared to infection with a non-aggregated one or multiple single bacilli in human MDMs [138]. Overall, the characterization of the microbiological factors important for the growth and survival of *M.tb* continues to shed new light on the host-pathogen interactions and deepen our understanding of bacterial virulence mechanisms.

## 9. Immune Factors: Cytokines

**TNF-α:** TNF-α is a pro-inflammatory cytokine important in controlling *M.tb* infection, and its production is primarily carried out by cells of the monocytic lineage, including macrophages [139]. During *M.tb* infection, TNF-α is one of the earliest cytokines to be produced. TNF-α signals through two trimeric membrane receptors: TNF receptors 1(TNF-R1) and (TNF-R2) [140] or by reverse signaling back into the membrane of TNF-producing cells [141]. Consequently, TNF signaling can result in a diversity of biological functions. Although TNF-α is considered to be a critical host resistance factor against TB, a recent report [43] showed that excess TNF confers susceptibility by increasing mitochondrial ROS, which initiates a signaling cascade to cause the pathogenic necrosis of mycobacterium-infected macrophages. In the context of *M.tb* infection, TNF-α is also involved in granuloma formation [142], and with the rise in the use of anti-inflammatory biologics like TNF-α inhibitors or antagonists, more recent studies have begun to characterize their roles in the reactivation of latent TB infection (LTBI) [143]. It was recently observed in human monocytes that TNF-α antagonists differentially induced the TGF-β1-dependent resuscitation of dormant *M.tb* [44]. A recent study suggests that CD153, a TNF super family member, plays an important role in *M.tb* control. *M.tb*-specific CD4^+^ T cells expressing CD153 were significantly reduced in patients with active TB when compared to those with LTBI [45].

**IL-6:** Interleukin-6 (IL-6) is a cytokine produced by several cell types, including macrophages, and is characterized to have both pro-inflammatory and anti-inflammatory functions [144,145]. Additionally, IL-6 is involved in processes such as acute phase response, cell growth and differentiation, and metabolic functions [146,147]. Given the pleotropic nature of IL-6, its role in TB infection remains less clear. For example, IL-6-deficient mice were highly susceptible to *M.tb* infection [46]. On the other hand, there is also evidence in the mouse model which suggests that IL-6 secreted by *M.tb*-infected macrophages inhibits the response of uninfected macrophages to IFN-γ [47]. A recent mouse study reported a novel IL-6 signaling mechanism where myeloid cell-like transcript 2 (TLT2) promotes IL-6 expression through the activation of STAT3 and the blocking of TLT2 results in a decreased bacterial load [148]. Further mechanistic studies also show the *M.tb* virulence factor, Rv3246c, enhances bacterial survival in macrophages by inhibiting TNF-α and IL-6 production in an NF-κB pathway-dependent manner [48]. To date, there are a limited number of reports and explanations on the role IL-6 plays directly in the immune response to TB. More experimentation is needed to fully understand its role, particularly in the context of human macrophages.

**GM-CSF:** Granulocyte-macrophage colony-stimulating factor (GM-CSF) has a well-characterized role in myelopoiesis. More recently, it has generated interest for its role in tissue inflammation and *M**.tb* infection [149]. A recent study reported GM-CSF to have the ability to skew macrophages to an M1 phenotype and effectively secrete pro-inflammatory cytokines [150]. In vitro studies show that targeting GM-CSF via monoclonal antibodies results in decreased IL-12p40 and TNF-α production upon BCG infection in mouse BMDMs [151]. Additionally, in both mouse and human macrophages, GM-CSF enhances *M.tb* localization in acidic compartments, resulting in phagolysosomal fusion and bacterial clearance [49,50]. Interestingly, following *M.tb* infection, human alveolar macrophages produce more GM-CSF conferring to them a higher capability to control infection when compared to mouse macrophages [51]. Additionally, blocking GM-CSF in human MDMs allows for an increase in bacterial growth, which further highlights a role for GM-CSF in bacterial control [49]. In the mouse model, the neutralization of GM-CSF in TNF-α-deficient mice with suboptimal isoniazid/rifampin treatment impairs the host inflammatory response and consequently leads to a high number of intracellular *M.tb* bacilli [151]. Given the recent findings and novel aspects of GM-CSF function, more attention should be given to its potential host-directed therapeutic applications.

**IL-1β:** The pro-inflammatory cytokine IL-1β is believed to play an important role in protecting the host against *M.tb* infection. Early studies with IL-1β-deficient mice showed that its absence leads to mycobacterial outgrowth in the lungs and distant organs, impaired granuloma formation, and a lack of Th1-mediated immune response [52]. Additionally, in the mouse model, the absence of the IL-1R signal leads to a dramatic early defect in the early control of *M.tb* infection due to an absence of MyD88-dependent signaling [53]. In a recent study, IL-1β was linked to limiting *M.tb* growth in mouse alveolar macrophages since its absence leads to a delay in the activation of the Th1 response [152]. Another recent study characterized a protective mechanism in which mice treated with β-glucan showed protection against pulmonary *M.tb*, and the mechanism was thought to be mediated via IL-1β signaling [153]. The interplay between *M.tb*-infected macrophages and macrophage metabolic processes has been tied to IL-1β since the anti-inflammatory microRNA-21 (miR-21), produced in response to proliferating mycobacteria, dampens glycolysis and causes the downregulation of IL-1β via an immunometabolic mechanism in mouse BMDMs [154]. A clinical study showed that macrophages with specific IL-1β responses differed between LTBI cases and active TB patients. Both IL-1β gene and protein expression were decreased in active TB patients when compared to LTBI subjects suggesting a potential role for IL-1β in preventing TB reactivation [54]. *M.tb* has evolved with adaptive capabilities to promote survival. A recent study characterizes a mechanism in which modern lineages of *M.tb* produce more IL-1β when compared to isolates of ancient lineages, thereby promoting IL-1β-induced autophagy, which is paradoxically associated with a high rate of intracellular bacilli replication [155].

**Type I Interferons:** The type I interferon family is composed of IFN-α (13 subtypes) and IFN-β, and these Interferons are classically associated with host defense in viral infections [156]. A role for type I IFNs in TB infection was established in a transcriptomic study of active TB patients, where the TB signature was dominated by a neutrophil-driven IFN-inducible gene profile, consisting of both IFN-γ and type I IFN-αβ signaling [157]. Since then, studies in the mouse model predominantly propose that type I IFNs play a detrimental role during TB infection. A recent study showed that the congenic mouse strain, B6.Sst1, which carries the “super susceptibility to tuberculosis 1” region of mouse chromosome 1 from C3HeB/FeJ mice on an otherwise B6 genetic background [158], was more susceptible to *M.tb* due to the uncontrolled production of type I IFN [55]. Interestingly, lung lesions in congenic sst1-susceptible mice show extensive necrosis and the unrestricted extracellular multiplication of *M.tb* [159]. In murine BMDMs, IFN-β signaling promotes host protection against *M.tb* infection by increasing the production of NO [58]. A recent study in murine BMDMs reports that type I IFN signaling correlates with decreased glycolysis and mitochondrial damage induced by *M.tb*, and the absence of type I IFN signaling allows for glycolytic flux and mitochondrial dysfunction both in in vivo and in vitro *M.tb* infections in macrophages [160]. Additionally, in the mouse model, the absence of type I IFN signaling in combination with Rifampin treatment leads to a significant reduction in CFUs in lungs and liver when compared to Rifampin alone [56]. In humans, several studies suggest that patients with active TB have increased levels of type I IFN and that this correlates with disease severity and poor clinical outcomes [57]. However, there are also clinical reports indicating a protective role of type I IFN. For example, type I IFN co-administration with antimycobacterial chemotherapy has been utilized clinically to treat MDR-TB, resulting in the improvement of clinical symptoms with a reduced bacterial burden [161]. Given the evolving mechanistic and clinical contradictory data, more research is needed to further characterize type I IFN-mediated signaling cascades that appear to be context-and species-dependent. 

**IFN-γ:** Interferon-γ (IFN-γ) is a type II Interferon and is primarily produced by CD4+ and CD8+ T cells [162]. It has long been thought that IFN-γ plays an important role in host defense against *M.tb* and nontuberculous mycobacterial pathogens by activating macrophages [162]. IFN-γ is an integral part of antibacterial signaling activities such as granuloma formation and phagosome-lysosome fusion, both of which lead to the control of intracellular mycobacteria. In human cells, reduced IFN-γ production is a marker of severe TB disease and is also utilized for the detection of *M.tb* infection [163]. Furthermore, IFN-γ has been linked with limiting intracellular bacterial replication by reducing hepcidin secretion in THP1 cells [59]. There is also recent evidence to suggest that IFN-γ regulates metabolic function by promoting glycolysis through inhibiting microRNA 21 (miR-21) in mouse BMDMs during *M.tb* infection [154]. Additionally, mouse models show that IFN-γ depletes intracellular histidine, which is essential for *M.tb* survival [60]. Research continues on the important roles that IFN-γ plays in limiting *M.tb* growth in macrophages.

**IL-10:** Interleukin 10 (IL-10) was originally characterized as a chemokine with predominantly anti-inflammatory properties that impede pathogen clearance by inhibiting Th1 cells, NK cells, and macrophages [164]. During *M.tb* infection, IL-10 production in macrophages is upregulated via the TLR2-ERK pathway [61]. The binding of IL-10 to its receptor activates a major JAK1-TYK2-STAT3 signaling cascade, which promotes the induction of the STAT3-mediated anti-inflammatory response, diminishing antibacterial activity [62]. Additionally, several mouse studies have characterized IL-10 activity, which allows *M.tb* to evade the host immune response and suppress macrophage function [165]. IL-10 has been previously targeted as a host-directed therapy via the IL-10-STAT3 pathway [166]. Furthermore, in *M.tb*-infected human macrophages, IL-10 has been shown to block phagosome maturation [63], resulting in increased bacterial survival. In contrast, evidence from an in vitro study shows that alveolar lining fluid (ALF) in culture promotes the release of *M.tb* cell wall fragments, resulting in the production of IL-10, which, coupled with STAT3 signaling, leads to the macrophage-mediated control of *M.tb* growth [64]. These contradictory results provide clues for how variables such as environment, location, and timing influence IL-10 signaling and its downstream consequences on infection biology.

**TGF-β:** TGF-β, produced by monocytes, has been well characterized as a driver of tissue repair. TGF-β also induces oxidative stress and promotes cell death [167]. A model has been recently reported that the presence of TGF-β in TB granulomas inhibits the killing of infected macrophages by cytotoxic T cells [65]. Additionally, in vitro granuloma studies have established that Adalimumab (ADA), an anti–TNF-α–targeting molecule, specifically mediates the TGF-β1-dependent resuscitation of dormant-like *M.tb* [44]. However, further delineation of the underlying mechanisms for TGF-β’s activities is needed. 

**IL-12:** IL-12 is produced by dendritic cells and macrophages in response to *M.tb* [168]. Several studies propose a central role of IL-12 in mounting an immune response and the intracellular killing of pathogens. It has been reported that IL-12p70 promotes macrophage bactericidal activity, proliferation, and cytosolic activity [66]. Several studies have reported that mutations in IL-12 confer increased susceptibility to *M.tb* infection [67]. Another study characterized mycobacterial genetic mutations leading to the modification of cell wall mycolic acids that results in the enhancement of IL-12 release by macrophages [169]. Given its central role in host defense, it makes sense that *M.tb* has developed immune evasion strategies that target IL-12 function.

## 10. Immune Factors: Chemokines

Chemokines play an important role in orchestrating the recruitment of cells including macrophages into the *M.tb*-infected lung, which contributes to *M.tb* containment but also can harm the host by contributing to inflammation and cavitation in the lungs during disease progression [68]. Plasma chemokines can be used as biomarkers of disease severity, higher bacterial burden, and delayed sputum culture conversion in pulmonary TB [170]. A plasma chemokine signature can also be used as a novel biomarker for predicting adverse treatment outcomes in pulmonary TB [171]. A recent study [69] identified that the baseline levels of plasma chemokines CCL1, CCL3, CXCL1, CXCL2, and CXCL10 were significantly higher in active TB (both microbiologically confirmed and clinically diagnosed TB) in comparison to TB controls in children. Consequently, these findings suggest that these chemokine signatures could also serve as biomarkers for the diagnosis of pediatric TB [69].

## 11. Host Genetic Factors

Previously, we and others have reviewed polymorphisms in genes of a wide variety of host innate immune factors associated with TB risk or resistance [172,173]. Here, we focus only on recent reports of selected classes in which host innate immune factors have been linked to susceptibility to or protection from TB (Table 2).

### 11.1. Cytokines

Changes in the production and release of cytokines affecting macrophage function are often linked to TB susceptibility or protection. For example, a systematic meta-analysis exploring the association between IFN-γ polymorphisms and the risk of developing TB found that there is an association between IFNG-γ +874 T/A (rs2430561) and the development of disease [174]. Another study found that single nucleotide polymorphisms (SNPs) in TNF were associated with protection against TB due to increased TNF expression and secretion [175]. A cohort study reported SNPs in TNF, IL-6, and IL-1β that possibly affected the levels of these cytokines and led to susceptibility or protection against TB infection [175]. An SNP in IL-17 that resulted in decreased IL-17A production upon stimulation was associated with TB susceptibility in a Chinese Han population [176]. Interestingly, a study that examined the master regulators of IL-12 and IL-10 signaling found that variants resulting in the increased production of IL-12 were associated with susceptibility to TB but an SNP that resulted in increased IL-10 production was associated with decreased pediatric TB [177]. In conclusion, variations in cytokine responses resulting from their genetic polymorphisms can lead to the enhanced or decreased control of TB infection, although it is important to note that too much of a cytokine can also be as detrimental as too little. Linking these polymorphisms specifically to defects in macrophage immune response awaits further studies.

### 11.2. Receptors

Differences in the activities of the cellular receptors, especially of macrophages, that recognize *M.tb* for uptake and that are associated with TB pathogenesis have also been identified. A case-control study of a Chinese Han cohort identified SNPs in PRRs, TLR8, and TLR9, which were found to be associated with the development of TB [178]. In another study with a Han Taiwanese population [179], the C-T haplotype in the TLR2 gene, in comparison with the most common T-T haplotype, was associated with an increased risk for TB. There is evidence to justify further evaluation of TLR2 polymorphisms and their effect on TLR2-dependent signaling as these findings can provide insight into the development of TB therapeutics [195]. Mutations in NOD2, an intracellular PRR that recognizes TB, were also identified and associated with increased susceptibility [180]. Studies on another PRR, Mincle or CLEC4E, have also been conducted and the results suggest the influence of ethnicity on the presence of significant polymorphisms in this receptor [181,196]. SNPs in MARCO and CD36, two scavenger receptors, have been found related to pulmonary TB risk [182]. Lastly, although not a receptor involved in the phagocytosis or uptake of *M.tb*, multiple recent studies have examined polymorphisms in the vitamin D receptor (VDR) in different populations for their association with increased susceptibility to TB infection [183,184,185,197]. Other important immune factors in which SNPs have been recently studied include NLRP3 [186], AIM2 [179,187], and C-type lectins [188,189].

### 11.3. Collectins (SP-A and SP-D)

Collectins are collagen-containing calcium-dependent (C-type) lectins that function to assist in the clearance of pathogens and particles in the lungs. Surfactant proteins A (SP-A) and D (SP-D) are collectins that are expressed by alveolar type II epithelial cells. Both SP-A and SP-D have been shown to be involved in regulating the phagocytosis of *M.tb* by macrophages [198,199]. Additionally, SP-D has been shown to reduce the intracellular growth of *M.tb* in macrophages [200]. A case-control study employing an in vitro assay illustrated that an *SP-D* polymorphism had the lower binding ability and less inhibition of the intracellular growth of *M. bovis* BCG in a murine alveolar macrophage cell line (MH-S cells) [190]. A measurement of allelic mRNA expression imbalance in collectins and several C-type lectin genes from human lung tissues revealed a frequent regulatory SNP in the SP-A gene in our own study [201]. In the Chinese Han population, both disease risk and protective correlations were reported between the presence of SNPs in the SP-A gene and pulmonary TB [191]. Apart from genetic polymorphisms, there is evidence to support that aging plays a role as macrophages infected with *M.tb* that had been exposed to alveolar lung fluid from elderly individuals (E-ALF) are less capable of controlling *M.tb* due to dysfunctional SP-A and SP-D in the E-ALF [202]. Similarly, mice infected with *M.tb* exposed to E-ALF displayed a significantly higher bacterial burden in the lung [202].

### 11.4. NRAMP1

The Natural Resistance-Associated Macrophage Protein 1 (NRAMP1) gene produces an integral membrane protein that functions as a divalent ion channel for transporting metals [203]. This protein is expressed exclusively in the lysosomal compartment of monocytes and macrophages and, after phagocytosis, NRAMP1 is targeted to the microbe-containing phagosomal membrane where it controls intracellular microbial replication by actively removing iron or other divalent cations from the phagosomal site [204]. Several studies have characterized the association of NRAMP1 with TB susceptibility in different parts of the world [192,193]. Recently, a case-control study in India analyzing the effects of a polymorphism in the NRAMP1 gene (3′UTR) showed an increase in host susceptibility to TB [194].

Overall, more follow-up work remains to be performed on elucidating the mechanisms by which genetic variants in innate immune factors, especially with regard to macrophages, result in protection or susceptibility to TB and how this can broaden our understanding of the interplay between host and pathogen.

## 12. Conclusions

TB remains one of the most lethal infectious diseases in the world. There has been a surge of research on cellular and biochemical pathways that are important in advancing our understanding of how *M.tb* has evolved to thrive in its intracellular niche, macrophages. The previously unknown roles of various novel proteins and small molecules from both host and pathogen in TB infection have also recently been described. In this review, we discuss several recent host innate immune macrophage factors as well as microbiological factors that play crucial roles during in vitro and in vivo infection by *M.tb* in either controlling or promoting bacterial survival in macrophages. Given the importance of innate immune factors like cytokines and bioeffector molecules in the host response to TB, work continues on characterizing the effects of their signaling cascades on the inflammatory response and its resolution. Mutations or SNPs in the genes for cytokines and other important molecules continue to be described, although much work remains to uncover the functional mechanisms by which these SNPs exert their effects. Besides variations in host genetics, differences in infecting *M.tb* strains and other microbiological factors that influence *M.tb* fitness offer additional insight into the pathogenesis and response of TB. Since TB infection remains a global public health problem in an era of increasing antibiotic resistance, we contend that a more comprehensive understanding of different immune factors that promote bacterial growth or protect against *M.tb* growth in macrophages will allow for the development of novel approaches for host-directed therapies that augment the activity of antibiotics against this deadly disease.

## Figures and Tables

**Table 1 pathogens-11-01153-t001:** Immune Factors Regulating *M.tb* Infection of Macrophages.

Innate Factors	Host/Cell Type	Biological Effects during Host-*M.tb* Interaction	References
**Cellular Pathways:**			
Mcl-1	Human MDMs	Treatment of macrophages with Mcl-1 antagonists resulted in significantly decreased *M.tb* growth	[8]
Caspase-8	Mouse macrophages	Drives cell death of *M.tb*-infected macrophages, thereby controlling infection	[9]
Sirtuin 7	Mouse RAW 264.7	Helps control *M.tb* growth through NO-induced apoptosis	[10]
mTOR	Mice	Increases autophagy and controls *M.tb* growth in infected mice (lung homogenate)	[11]
Hydrogen sulfide	Mouse RAW 264.7	Increases autophagy and controls *M.tb* growth in infected cells	[12]
HIF-1	Human U937 monocytes	Increases autophagy and controls *M.tb* growth in infected cells	[13]
DRAM2	Human MDMs	Binds to microtubule-associated proteins essential for the initiation of autophagy and decreases *M.tb* growth	[14]
microRNA miR-18a	Mouse RAW 264.7	Decreases LC3-II expression, required for autophagosome formation, and promotes *M.tb* survival	[15]
CLEC4E	Mouse BMDMs	Enhances autophagy and decreases *M.tb* growth	[16]
TLR4	Mouse BMDMs	Enhances autophagy and decreases *M.tb* growth	[16]
Sirtuin 3	Mouse BMDMs	Involved in the expression of PPARα, an autophagy activator, and its KO macrophages show increased growth of *M.tb*	[17]
P2X7	Mice	Detects ATP released during cellular stress or death pathways and activates the inflammasome, leading to decreased disease severity and *M.tb* CFUs in lung	[18]
P2RX7	Zebrafish	Potentiation through the drug clemastine improves mycobacterial infection control	[19]
HIF-1α	Human MDMs	Presence in normoxic conditions decreases intracellular *M.tb* growth and also decreases the release of TNFα and IL-10.	[20]
	Mice	Deficiency of HIF-1α increases lung bacterial burden in infected mice	[21]
	Mouse BMDMs	Deficiency of HIF-1α increases *M.tb* growth and impairs the expression of glycolysis-related genes	[22]
**Bioeffector Proteins and Molecules:**			
TLR2	Mouse BMDMs	A late, endosome-specific component of the TLR2 response is inhibited by *M.tb* virulence factors PDIM and ESX-1 to improve *M.tb* growth.	[23]
	Mice	Critical for activation of Sirtuin 3 and protection against *M.tb* in macrophages from lung and spleen	[24]
TLR9	Mouse BMDMs	Recognizes unmethylated CpG motifs in bacterial DNA and plays role in the recognition and control of *M.tb* infection	[25]
IDO-1	Human and mouse macrophages	Expression is upregulated by *M.tb* infection but is not essential for the control of *M.tb* growth in vitro.	[26]
	Mice, Non-human primates (NHP)	Expression correlates with the increase in mouse lung CFUs of *M.tb* and treatment of rhesus macaques with an IDO-1 inhibitor decreases lung bacterial burden.	[27]
	Human PBMCs, MDMs	Represents one of the biochemical pathways in human macrophages that prevents the efficient killing of *M.tb* in TB granulomas	[28]
FAK	Human THP-1	Overexpression leads to decreased *M.tb* survival, which is due to increased ROS production	[29]
microRNA miR-495	Human THP-1	Causes decreased *M.tb* survival through the increased production of ROS and inhibition of SOD2	[30]
TARM-1	Mouse RAW 264.7	Knocking down of this receptor decreases the production of ROS and increases the growth of *M.tb* H_37_R_v_	[31]
FAO	Mouse BMDMs	Inhibition of fatty acid oxidase leads to NADPH oxidase recruitment and decreased *M.tb* growth	[32]
CD157	Human MDMs	Macrophage treatment with soluble CD157 leads to decreased CFUs of *M.tb*, likely due to TLR2-dependent ROS production	[33]
Liposomal glutathione	Mice	An antioxidant that prevents damage to host immune cells by ROS, but decreases lung CFUs of *M.tb* at the same time	[34]
Vitamin D	Mice	Activated form induces the synthesis of LL-37, and administration of which in *M.tb*-infected mice leads to a reduction in lung bacterial burden	[35,36]
Vitamin D + Phenylbutyrate	Human MDMs	Inhibit the growth of MDR-TB strains	[37]
PPARα	Mouse BMDMs	Protective against infection, since KO mice show increased *M.tb* growth compared to WT	[38]
PPARy	Human MDMs	Permissive to infection, since knocking down of the gene significantly decreases *M.tb* growth concomitant with an increase in TNF	[39]
**Other Cellular Factors:**			
Airway epithelial cells (AECs)	Human alveolar epithelial cells	Express PRRs, surfactant, and recruit neutrophils. Provide protective host response against *M.tb* infection in the airway environment which contains alveolar macrophages.	[40,41]
	Lung-on-chip model with mouse cells	Cells lacking surfactant allowed for the rapid growth of *M.tb* further highlighting the importance of surfactant in bacterial control in the alveolar environment which contains alveolar macrophages	[42]
**Cytokines and Chemokines:**			
TNF-α	Zebrafish macrophages	Considered as a critical host resistance factor against TB but excess TNF confers TB susceptibility by increasing mitochondrial ROS in infected macrophages.	[43]
	Human PBMCs/ in vitro granuloma	TNF-α antagonists differentially induce TGF-β1-dependent resuscitation of dormant *M.tb*	[44]
CD153	Human T cells	*M.tb*-specific CD4^+^ T cells expressing CD153 is significantly reduced in patients with active TB	[45]
IL-6	Mice	KO mice are highly susceptible to *M.tb* infection	[46]
	Mouse macrophages	*M.tb*-induced IL-6 inhibits macrophage response to IFN-γ	[47]
	Human U937, Mouse RAW 264.7	*M.tb* virulence factor, Rv3246c, enhances bacterial survival in macrophages by inhibiting TNF-α and IL-6 production in an NF-κB pathway-dependent manner	[48]
GM-CSF	Human MDMs, Mouse AMs and RAW 264.7	Enhances *M.tb* localization in acidic compartments, resulting in phagolysosomal fusion and bacterial clearance	[49,50]
	Human and Mouse AMs	Human AMs after *M.tb* infection produce more GM-CSF, conferring higher control of infection than mouse AMs	[51]
IL-1β	Mice	Absence leads to *M.tb* outgrowth in the lungs and distant organs and impaired granuloma formation containing fewer macrophages	[52]
	Mice, Mouse macrophages	Absence of IL-1R signal leads to a dramatic defect in early control of *M.tb* infection in vivo and also in stimulated macrophages due to an absence of MyD88-dependent signaling	[53]
	Human MDMs	Both gene and protein expression are decreased in MDMs from active TB patients compared to LTBI subjects, suggesting a role for IL-1β in preventing TB reactivation	[54]
Type I IFN	Mice (B6.Sst1 strain)	Uncontrolled production of type I IFN by mice increases their susceptibility to *M.tb*	[55]
	Mice, Mouse BMDMs and RAW 264.7	Type I IFN signaling mediates *M.tb*-induced macrophage death. Its absence in combination with Rifampin treatment leads to a significant reduction in CFUs in lungs and liver	[56]
	Human	Patients with active TB have increased levels of type I IFN, which correlates with disease severity	[57]
IFN-β	Mouse BMDM	IFN-β signaling promotes protection against *M.tb* infection by increasing the production of NO	[58]
IFN-γ	Human THP-1	Associated with limiting intracellular bacterial replication by reducing hepcidin secretion in macrophages	[59]
	Mice, Mouse macrophages	Depletes intracellular histidine which is essential for *M.tb* survival both in vivo and ex vivo murine macrophages	[60]
IL-10	Human and mouse macrophages	Production in macrophages is upregulated via the TLR2-ERK pathway during *M.tb* infection. IL-10, through STAT3 induction, mediates anti-inflammatory response and diminishes antibacterial activity	[61,62]
	Human AMs, MDMs and THP-1	Blocks phagosome maturation resulting in increased *M.tb* survival in macrophages	[63]
	Human MDMs	Presence of alveolar lining fluid in infected cell culture releases *M.tb* cell wall fragments resulting in the production of IL-10 which, coupled with STAT3 signaling, leads to macrophage-mediated control and growth of *M.tb*	[64]
TGF-β	*GranSim* granuloma model, NHP	Presence of TGF-β in TB granulomas inhibits the killing of infected macrophages by cytotoxic T cells	[65]
	Human PBMCs/ in vitro granuloma	Adalimumab, an anti–TNF-α–targeting molecule, specifically mediates TGF-β1-dependent resuscitation of dormant *M.tb*	[44]
IL-12	Human PBMCs, macrophages	Promotes macrophage bactericidal activity, proliferation, and cytosolic activity	[66]
	Human/Mouse macrophages and DCs	Mutations in IL-12 confer increased susceptibility to *M.tb* infection	[67]
Chemokines	Human immune cells, macrophages	Recruit cells including macrophages into the *M.tb*-infected lung which contributes to *M.tb* containment	[68]
CCL1, CCL3, CXCL1, CXCL2, CXCL10	Confirmed and Control TB subjects	Baseline levels of these plasma chemokines are significantly higher in active TB patients compared to TB controls in children	[69]

**Table 2 pathogens-11-01153-t002:** Innate Immune Gene Polymorphisms that Affect Macrophage Response and TB Disease in Populations.

Gene	Polymorphism	Population	Association with TB	Effect	References
IFN-γ	+874 T/A (rs2430561)	American, European, African, Asian	Yes	Susceptible	[174]
TNF	rs1799964, rs1800630	Chinese Tibetan	Yes	Susceptible	[175]
	rs1799724, rs1800629	Tibetan	Yes	Susceptible	[175]
IL-1β	rs16944	Chinese Han, Tibetan	Yes	Protective	[175]
IL-6	rs2069837	Chinese Han	Yes	Protective	[175]
IL-17A	rs8193036	Chinese Han	Yes	Susceptible	[176]
REL/IL-12	rs842618	Vietnamese	Yes	Susceptible	[177]
BHLHE40/ IL-10	rs11130215	South Africa	Yes	Protective	[177]
TLR8	rs3764880	Chinese Han	Yes	Protective	[178]
TLR9	rs187084	Chinese Han	Yes	Susceptible	[178]
TLR2	rs3804099, rs3804100	Han Taiwanese	Yes	Susceptible	[179]
NOD2	rs1861759, rs7194886	Chinese Han	Yes	Susceptible	[180]
	rs2066842, rs2066844	African Americans	Yes	Protective	[180]
CD14	rs2569190, rs2569191	Chinese Han	Yes	Susceptible	[180]
Mincle (CLEC4E)	rs10841847	West African	Yes	Susceptible	[181]
MARCO	rs12998782	Chinese Han	Yes	Susceptible	[182]
CD36	rs1194182, rs10499859	Chinese Han	Yes	Protective	[182]
VDR	Fok1(T/C) genotype	North Indian	Yes	Susceptible	[183]
	Folk1(ff) genotype	Chinese	Yes	Susceptible	[184]
	BsmI (rs1544410) TaqI (rs731236),	Iranian	Yes	Susceptible	[185]
CARD8	rs2043211	Ethiopian	Yes	Susceptible	[186]
NLRP3	rs35829419	Ethiopian	Yes	Susceptible	[186]
AIM2	rs1103577	Brazilian	Yes	Protective	[187]
MASP1	rs3774275	Indian	Yes	Protective	[188]
MBL2	A/O, O/O genotype	Polish	Yes	Susceptible	[189]
SP-D	rs721917	Taiwanese	Yes	Susceptible	[190]
SP-A	rs17886395, rs1965707	Chinese Han	Yes	Susceptible	[191]
NRAMP	5′(CA)_n_, INT4, D543N, and 3'UTR	African, American	Yes	Susceptible	[192,193]
	3'UTR	Indian	Yes	Susceptible	[194]

## Data Availability

Not applicable.

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
