# Peer review of "Resistance and Susceptibility Immune Factors at Play during Mycobacterium tuberculosis Infection of Macrophages"

_pathogens, 2022, doi:10.3390/pathogens11101153_

Round 1

Reviewer 1 Report

The authors beautifully review the factors that have a role in TB infection. Especially focusing on macrophages that are the main host for this intracellular pathogen. The article is well written and structured, and provides the reader with useful tables that achieve to summarize the complexity of reviewed literature. It is really appreciated to have new concepts/factors briefly introduced before getting into the details of their effects on TB immune responses.

Introduction:

Line 46: please add a reference.

Table1:  Suggestion to merge cells of the first column when the factor has different effects depending on the Host/Cell type. Example: HIF-1a, merge the 3 cells in the first column.

Microbiological factors:

Some emphasis on the lipid profile differences among TB lineages would be relevant to be discussed in this section.

Table 2: same suggestion as for table 1. Also consider eliminating column “association with TB”, if there is any association that’s stated in column “effect”.

There are some typos or formatting errors that need to be addressed for instance in lines: 219, 226, 291, 430, 435, 442, 489, 505, and 532-33.

Reviewer 2 Report

In this manuscript titled “Resistance and Susceptibility Immune Factors at play during Mycobacterium tuberculosis Infection of Macrophages”, the author did a thorough review of the immune components which play roles in Mycobacterium infections. The authors nicely summarized the recent advances on this topic, and clearly distinguished the findings on mice and human. Overall, this is an interesting and important manuscript. And it is well written for most parts. I only have several minor questions.

Minor issues:

1. The M. tb stains could be recognized by multiple host immune receptors, especially those on macrophages, and they can also partially counter some of the recognition. Therefore, it seems important to briefly introduce how M. tb infections occur in hosts, which help the readers get a big picture of how these innate immune responses are initiated and utilized by M. tb.

2. It seems that different models were used for the studies on the interaction between M.tb and macrophages, such as different human macrophage cells, different animal models and different mouse cells. It may help the readers better understand these findings by briefly summarizing the models as well as pros and cons of these models. For examples, RAW 264.7 cells were mentioned multiple times in this manuscript, but it is never introduced what RAW 264.7 cells are.

3. I don’t see the difference between the section “Cellular Pathways” and section “Bioeffector Determinants”. Why can’t they be introduced in the same section? To me, they all seem like immunological signaling pathways in macrophages. At least, the authors need better names for these sections.

4. Line 217-219, the description is vague, details are needed.

5. Line 237-239, “perforation of phagosomes in macrophages is mediated by ESX-1 through ESAT-6, thereby enhancing M.tb virulence”. This description is vague, details are needed.

6. In table 2, I don’t know why “rs842634” is listed in this table if this polymorphism is not associated with TB. Actually, the authors didn’t introduce it in the main text either. It seems that this specific polymorphism is irrelevant with the topic of this manuscript.
